# One-Step Synthesized Iron-Carbon Core-Shell Nanoparticles to Activate Persulfate for Effective Degradation of Tetrabromobisphenol A: Performance and Activation Mechanism

**DOI:** 10.3390/nano12244483

**Published:** 2022-12-18

**Authors:** Yunjiang Yu, Chang Liu, Chenyu Yang, Yang Yu, Lun Lu, Ruixue Ma, Liangzhong Li

**Affiliations:** 1State Environmental Protection Key Laboratory of Environmental Pollution Health Risk Assessment, Center for Environmental Health Research, South China Institute of Environmental Sciences, Ministry of Ecology and Environment, Guangzhou 510655, China; 2Inner Mongolia Autonomous Region Key Laboratory of Water Pollution Control, School of Ecology and Environment, Inner Mongolia University, Hohhot 010021, China; 3Guangdong Key Laboratory of Environmental Pollution and Health, School of Environment Jinan University, Guangzhou 511443, China

**Keywords:** iron-carbon core-shell nanoparticle, synergistic catalysis, tetrabromobisphenol A, mechanism

## Abstract

Tetrabromobisphenol A (TBBPA), as an emerging endocrine disrupter, has been considered one of the persistent organic contaminants in water. It is urgently necessary to develop an efficient technique for the effective removal of TBBPA from water. Herein, a one-step hydrothermal synthesis route was employed to prepare a novel iron-carbon core-shell nanoparticle (Fe@MC) for effectively activating persulfate (PS) to degrade TBBPA. Morphological and structural characterization indicated that the prepared Fe@MC had a typical core-shell structure composed of a 5 nm thick graphene-like carbon shell and a multi-valence iron core. It can be seen that 94.9% of TBBPA (10 mg/L) could be degraded within 30 min at pH = 7. This excellent catalytic activity was attributed to the synergistic effect of the porous carbon shell and a multi-valence iron core. The porous carbon shell could effectively prevent the leaching of metal ions and facilitate PS activation due to its electron transfer capability. Furthermore, numerous micro-reaction zones could be formed on the surface of Fe@MC during the rapid TBBPA removal process. Radical quenching experiments and electron paramagnetic resonance (EPR) technology indicated that reactive oxygen species (ROS), including OH, SO_4_^−^, O_2_^−^**,** and ^1^O_2_, were involved in the TBBPA degradation process. Based on density functional theory (DFT) calculation, the carbon atoms linked by phenolic hydroxyl groups would be more vulnerable to attack by electron-rich groups; the central carbon was cracked and hydroxylated to generate short-chain aliphatic acids. The toxicity evaluation provides clear evidence for the promising application potential of our prepared material for the efficient removal of TBBPA from water.

## 1. Introduction

Tetrabromobisphenol A (TBBPA), a typical brominated flame retardant, was widely used to produce plastics, electronic products, construction materials, and textiles [1] (Li, et al. 2020). Due to its widespread use and high durability, TBBPA has been detected in various water environments such as lakes, rivers, and oceans. Moreover, it has been reported that TBBPA can pose a severe risk to human health owing to its high toxicity to hepato-kidney, immune, and nervous systems and endocrine disrupting effects [2,3,4]. Therefore, it is necessary and urgent to develop an efficient technology for effectively removing TBBPA from water.

In recent years, sulfate radical-based advanced oxidation processes (AOPs) have attracted increasing attention in water treatment [5]. Through the activation of peroxymonosulfate (PMS) and persulfate (PS) via heat treatment [6], photoactivation [7,8], and transition metal (e.g., iron, cobalt) activation [9,10], abundant free radicals can be generated for the effective decomposition of refractory organic pollutants from water.

Compared with other free radicals, SO_4_^−^ has higher redox potential (E_0_ = 2.5–3.1 eV), longer half-life (30–40 μs), and a more robust selectivity for pollutants with aromatic benzene rings [11]. Zero-valent iron (ZVI), as a slow-release source of Fe(II), has been proven to be effective for the activation of persulfate.

However, the surface oxidation and agglomeration of ZVI nanoparticles are still the main obstacles to its widespread application [12]. In order to overcome the above-mentioned problems, carbon-based porous materials were used as the supporter to immobilize iron-based catalysts [13,14]. During the high-temperature synthesis procedure of the iron-carbon composite materials, the graphene-like structure can be formed to effectively prevent the agglomeration and leaching of iron [15]. For example, an encapsulated graphite layer embedded in mesoporous carbon (MCFe), which was prepared by the chelation-assisted assembly and carbothermal reaction, was able to more effectively prevent the metal leaching compared to other iron-based catalysts [16]. A nanocomposite catalyst (Fe@CNs) was reported as an efficient PS activator to degrade bisphenol A from water, attributing to the encapsulation of Fe nanoparticles in the carbon matrix under double protection of carbon shell and adherent graphite.

In this study, we used glucose as the carbon source and Fe^3+^ as the iron source to synthesize mesoporous iron-carbon catalyst (Fe@MC) with a typical core-shell structure by the hydrothermal method. Glucose, one of the most widely distributed and essential monosaccharides in nature, has become an attractive biocarbon procurer due to its lower requirement for operating pressure, temperature, and time than other biomasses (e.g., cellulose). Additionally, glucose can facilitate the formation of mesoporous carbon microspheres with a uniform size and large specific surface area. More importantly, glucose is a reducing sugar with polyhydroxy aldehydes and the ability to polymerize and carbonize. A series of techniques were used to investigate the structure of the prepared material. The radical quenching tests and electron paramagnetic resonance (EPR) technique were used to determine the primary radicals involved in the removal of TBBPA, and the degradation mechanism of TBBPA by the Fe@MC/PS system was proposed. Combined with the DFT calculation and LC-MS-MS analysis, the degradation pathway of TBBPA was illustrated. The comprehensive toxicity of the intermediate products was investigated as well.

## 2. Materials and Methods

### 2.1. Chemical and Materials

Tetrabromobisphenol A (TBBPA, C_15_H_12_Br_4_O_2_, 98%,), sodium hydroxide (NaOH, 97%), glucose (C_6_H_12_O_6_·H_2_O, 99%), iron(III) chloride hexahydrate (FeCl_3_·6H_2_O, 99.9%), sodium persulfate (Na_2_S_2_O_8_, 98.0%), ethanol (C_2_H_5_OH, 99.7%), tert-buty alcohol (C_4_H_10_O, 98.0%) were purchased from Anpu Experimental Technology Inc., Shanghai China. sodium carbonate (Na_2_CO_3_, 99.5%), sodium bicarbonate (NaHCO_3_, 99.5%), sodium acetate (NaAc, 99.5%), sodium dihydrogen phosphate dihydrate (NaH_2_PO_4_·2H_2_O, 99.5%), *p*-benzoquinone (BQ, 99.5%), *L*-histidine, and potassium iodide (KI, 99.5%) were purchased from Shanghai Macklin Biochemical Co. Ltd. (China), Shanghai, China.

### 2.2. Material Synthesis

A one-step hydrothermal reaction was employed to synthesize iron-carbon core-shell nanoparticles. In brief, 4 g glucose powder and 1.38 g FeCl_3_·6H_2_O were dissolved into 30 mL deionized (DI) water under magnetic stirring at room temperature. After adding 0.82 g NaAc, the mixture was transferred into a 100 mL teflon(PTFE) lined hydrothermal reactor at 160 °C for 6 h. The obtained powders were washed with DI water and ethanol to remove the unreacted reagents and then dried in a vacuum freeze dryer. Finally, the dried materials were calcinated in a tube furnace under a nitrogen atmosphere at a flow rate of 800 mL/min. The calcination temperature was gradually elevated to 500, 800, 1100, and 1300 °C at a heating rate of 10 °C/min, and the holding time of each step was 30 min. The synthetic route of Fe@MC is shown graphically in Figure 1.

### 2.3. Material Characterization

The crystal structure of the samples was analyzed by X-ray diffraction (XRD, SmartLab 3 KW, Rigaku Corporation Inc., Tokyo, Japan) in a 2*θ* range of 20–90° with a scanning rate of 10 °/min. Raman spectroscopy was performed at the laser wavelength of 532 nm (Renessau 2000 system, Renishaw Inc., UK). The surface morphology and structure of samples were investigated by a scanning electron microscope (SEM, S4800, Hitachi Inc., Tokyo, Japan) equipped with an energy dispersive spectrometer (EDS, 7593-H, HORIBA Inc., Tokyo, Japan) and a transmission electron microscope (TEM, Tecnai G20, FEI Inc. Portland, OR, USA). The surface element distribution and functional group composition of samples were detected by using X-ray photoelectron spectroscopy (XPS, ESCALAB 250XI, Thermo Fisher Inc. Waltham, MA, USA). The specific surface area and pore size distribution of samples were performed using a JW-BK 122 W static nitrogen adsorber (TriStar II 3020, McMurray Teck Inc., Norcross, GA, USA). The thermal stability of samples was evaluated by thermogravimetric analysis (TGA/DSC, Mettler Toledo Inc., Switzerland) in the temperature range of 25–800 °C at the heating rate of 10 °C/min under a nitrogen atmosphere.

### 2.4. Batch Experiments

The removal of TBBPA by the prepared material was carried out in a 250 mL Erlenmeyer flask. In brief, a certain amount of Fe@MC and PS was added into 100 mL TBBPA solution (10 mg·L^−1^) under stirring for 40 min. A solution of 0.1 M HNO_3_ and NaOH was added to adjust the initial pH value of the solutions. Samples were taken at different time intervals for measuring TBBPA concentration using high-performance liquid chromatography (Agilent 1260 HPLC, Agilent Technologies, Santa Clara, CA, USA). The effect of Fe@MC dosage (0.05–0.3 g·L^−1^), PS dosage (0–2 mM), solution pH (3–11), and co-existing substrates (Cl^−^, CO_3_^2−^ and HPO_4_^−^and natural organic matter (NOM)) on the removal performance was investigated by varying the corresponding reaction conditions. In the free radical quenching experiment, methanol (MeOH), tert-butanol (TBA), *p*-benzoquinone (BQ), *L*-histidine, and KI were added to quench the corresponding free radicals. After three cycles of catalytic reactions, the Fe@MC was collected and heated at 1100 °C under the nitrogen atmosphere to evaluate its reusability. According to the 1,10-phenanthroline method, the concentration of Fe(II)/(III) iron ions and total dissolved iron ions in the solution after the reaction was measured by using an ultraviolet/visible spectrophotometer at 510 nm. In addition, the UV/Visible spectrophotometer was used to quantify the concentration of bromide ions at 590 nm.

The removal efficiency was calculated as the below equation:(1)$$\mathrm{Removal}\;\mathrm{efficiency}\;\left(\%\right)=\frac{C_{0}-C\;}{C_{0}}$$where C_0_ is the initial concentration of TBBPA and C is the concentration of TBBPA at t time.

### 2.5. Identification of Transformation Products

The liquid chromatography (1260, Agilent Inc., shanghai, China) was used to detect different intermediate products at different reaction times; then the LC-MS-MS (6500, Agilent Inc., shanghai, China) was employed to identify the conversion products of TBBPA.

### 2.6. DFT Computational Details

All the calculations were performed in the framework of the density functional theory with the projector-augmented plane-wave method, as implemented in the Vienna ab initio simulation package [17]. The generalized gradient approximation (GGA) proposed by Perdew-Burke-Ernzerhof (PBE) was selected for the exchange-correlation potential [18]. Spin polarization was turned on for all calculations. The long-range van der Waals interaction is described by the DFT-D3 approach [19]. The cut-off energy for the plane wave was set to 400 eV. All atomic positions were fully relaxed until the forces were smaller than 0.02 eV/Å and the total energies converged to 10^−7^ eV. The unit cell dimensions were 16 Å × 16 Å × 16 Å for all calculations, large enough to ensure that there were no direct interactions between the original structure and its self-image within the periodic boundary conditions. All calculations were carried out at the *Γ* point.

### 2.7. Acute Toxicity Evaluation of Reaction Solutions

The solution at different reaction times was diluted 20 times with DI water, and then zebrafish embryos were cultured in a Petri dish containing the diluted solution, as mentioned above. We set the same volume of DI water as blank and set within-group and between-group controls. We placed these Petri dishes in a biochemical incubator at 27 ± 2 °C. Mortality and hatchability of zebrafish embryos (0–96 hpf) were observed periodically. Results were calculated using the mean determined from four parallel experiments (100 zebrafish embryos).

## 3. Results and Discussion

### 3.1. Performance of Materials Prepared with Different Conditions

It can be observed from Figure 2a that the hydrothermal temperature and time have no apparent influence on the catalytic property of the materials. The increase in hydrothermal time can enhance the thickness of the carbon shell on the Fe@MC catalyst surface, and thus reduce the contact probability between Fe active sites and PS. The Fe@MC catalysts prepared at a hydrothermal temperature of 160 °C for 10 h can achieve the best removal performance. Figure 2b shows that the removal performance could be improved by increasing the calcination temperature from 500 to 800 °C, achieving the elevation of TBBPA removal rate from 82.4% to 94.9%. At the higher calcination temperature, more Fe active sites could be formed on the Fe@MC for the decomposition of the TBBPA [15]. However, the removal efficiency dropped to 59.7% when the calcination temperature was further raised above 1000 °C. This might be due to the collapse of the surface structure of carbon materials at such high calcination temperatures, leading to the relative reduction of specific surface area. Based on the experimental results, the optimized Fe@MC catalyst was prepared at the hydrothermal temperature of 160 °C for 10 h, followed by a calcination treatment at 800 °C.

### 3.2. Optimal Removal of TBBPA

#### 3.2.1. Effect of Fe@MC Dosage, PS Dosage, and Initial pH

From Figure 2c, the removal efficiency and reaction rate constant of TBBPA is significantly increased by increasing the Fe@MC dosage from 5 mg/L to 10 mg/L and slightly decreased with the further increase in the Fe@MC dosage. This is because the excessive material dosage might cause the recombination reaction of the free radicals, resulting in the reduction of removal efficiency and reaction rate constant of TBBPA at the Fe@MC [20]. The reaction rate constant was based on constants acquired from the pseudo-first-order kinetics (ln(C/C_0_) = −kt, where C is the TBBPA concentration at a certain reaction time (*t*), and C_0_ is the initial TBBPA concentration, *k* is the reaction rate constant.

With the increase of PS dosage, Fe@MC can fully activate PS to generate enough free radicals for TBBPA removal (Appendix A). Like the Fe@MC dosage, the excessive PS in the solution also leads to the recombination reaction of the free radicals [21]. Therefore, the optimized PS dosage is 1 mM in this study.

Figure 2e shows the performance of the Fe@MC/PS system for TBBPA removal under different initial pH conditions. It can be seen that the removal efficiency of the Fe@MC/PS system on TBBPA remains above 85% within 30 min in the pH range of 5–9, and the highest removal of TBBPA was achieved at pH 7.0. This might be due to the presence of abundant oxygen-containing functional groups on the surface of the porous carbon layer in the Fe@MC that can provide a strong pH buffer effect. The apparent reduction in the removal efficiency of TBBPA under strongly acidic conditions (pH = 3) should be attributed to the fact that the excessive H^+^ ions in solution can stabilize S_2_O_8_^2−^ and then inhibit the PS activation process [22]. A more significant reduction in the removal efficiency of TBBPA is observed at pH 11, mainly attributed to the change of surface charge of Fe@MC material owing to the accumulation of OH^−^ ions. Moreover, TBBPA would be deprotonated at pH 11 and becomes dominant in the anionic form [23]. Electrostatic repulsion between the catalyst and S_2_O_8_^2−^ would be therefore enhanced in such a robust basic environment [24].

#### 3.2.2. Effect of Co-Existing Substrates

The catalysis property of Fe@MC/PS might be affected by co-existing anions (e.g., Cl^−^, HCO_3_^−^, and H_2_PO_4_^−^) and NOM in water. It can be seen from Figure 3a that the presence of Cl^-^ ions in water has a slight inhibitory effect on the removal of TBBPA. However, Figure 3b,c demonstrates that the addition of HCO_3_^−^ and H_2_PO_4_^−^ could significantly affect the removal of TBBPA by Fe@MC, leading to a reduction in removal efficiency of 36.5% and 19.2%, respectively. This is consistent with the previous studies that HCO_3_^−^ and H_2_PO_4_^−^ can usually react with SO_4_^−^ and act as the quenchers for free radicals in the advanced oxidation processes [25]. Although Cl^−^ ions may also react with SO_4_^−^, the formed Cl_2_^−^ radical (the redox potential of Cl_2_^−^ and SO_4_^−^ is, respectively, 2.1 V and 2.5–3.1 V) still has the oxidative activity to degrade TBBPA to some extent [26]. Herein, the inhibitory effect of Cl^−^ ions is much lower than that of HCO_3_^−^ and H_2_PO_4_^−^.

As shown in Figure 3d, the presence of humic acid (HA) as the representative of NOM in water can slightly influence the removal of TBBPA. As well known, HA, as hydrophobic organic matter, would be readily adsorbed on the carbon material surface via hydrophobic-hydrophobic interaction. The accumulated HA on the surface of Fe@MC as a typical non-conductive layer would inhibit the transfer of conductive electrons between the catalyst and PS [27]. In addition, HA can occupy the active sites on the Fe@MC catalyst to prevent the adsorption and oxidation of TBBPA during the removal process.

Besides, in order to testify to the practical applicability of the catalyst, actual contaminated water (influent of the sewage treatment plant) (Appendix A) was selected for TBBPA removal tests. Appendix A shows the effect on the degradation performance of TBBPA under different water quality conditions. In the actual contaminated water, the TBBPA removal rate was only 69.3%, probably due to the higher TOC concentration. The reactive oxygen species produced by the Fe@MC/PS system are consumed. Appendix A showed the PS dosage is proportional to the TBBPA removal effect; when the dosage of PS was 10 mM, 89.2% of TBBPA was removed, and the removal rate of TOC was 60.1%. The above results show that actual contaminated water would affect the removal of TBBPA; the increase of PS dosage also could promote TBBPA removal.

### 3.3. Stability and Reusability

The reusability of the catalyst should be an essential factor for its application potential in water treatment. As shown in Appendix A, Fe@MC demonstrates good reusability in the four cycles of removal experiments. The removal of TBBPA in the Fe@MC/PS system decreased by 10% in each cycle. After four cycles, the removal rate of TBBPA by the system was reduced to about 60%. The decrease in the number of active sites should be responsible for the reduction in TBBPA removal efficiency. After the four-cycle removal experiments, the Fe@MC material was calcined at 500 °C under the N_2_ atmosphere. The TBBPA degradation rate of the spent Fe@MC could recover to over 90%, indicating that the calcination treatment can effectively regenerate the catalysis performance of Fe@MC.

### 3.4. Mechanism Study

#### 3.4.1. Characterization of Materials

The surface morphology of Fe@MC was observed by SEM. As shown in Figure 4a, the Fe@MC core-shell material is aggregated by numerous particles with a diameter of several dozen nanometers (Appendix A). This might be due to the mutual attraction among the component particles via the carbon-based adhesion and magnetic properties of Fe@MC materials [28]. A typical core-shell structure of Fe@MC with an outer layer of 5 nm-thick graphene-like carbon and the inner core of mixed iron (Fe(0), Fe(II), Fe(III)) components can be observed from the TEM image of Fe@MC as shown in Figure 4b. The formed core-shell structure can not only increase the specific surface area of the Fe@MC material but also prevent the agglomeration and leaching of internal metals due to the presence of a graphene-like carbon outer layer and can facilitate the transportation of electrons during the reaction. Raman spectra were used to analyze the carbon structures of Fe@MC; the results are shown in Figure 4d. Three characteristic peaks detected at 1337, 1586, and 2676 cm^−1^ were, respectively, assigned to D, G, and 2D bands, indicating the presence of amorphous carbon and a graphite structure. The D band is associated with disordered vibrations of *sp^3^* carbon caused by the symmetry-fractured structural defects or edges [29]. Thus, the relative intensity of the D and G bands (I_D_/I_G_) can be used to identify carbonaceous structures. In this study, the I_D_/I_G_ ratio of Fe@MC (1.06) was more significant than other previously reported carbon-based materials (0.82), suggesting that thFe@MC may provide more defects for enhancing catalytic performance. The TGA curve (Appendix A) shows that the mass loss of Fe@MC was only 3.22% at the temperature range of 700 °C, lower than other previously reported carbon-based materials. The higher thermal stability of Fe@MC may have originated from the presence of relatively strong graphite (G) bands. Based on the analysis of XPS and EDS, elements including C, O, and Fe can be detected on the surface of Fe@MC with a relative content of 95.77%, 2.17%, and 2.06% (Appendix A). The detection of the Fe element indicates that Fe components have been successfully wrapped in the carbon shell. The core-level C1 peak can be further decomposed into five component peaks assigned to C-C/C-H (284.6 eV), C-O (285.9 eV), C=O (287.1 eV), O-C=O (288.9 eV) bonds and π-π* shake-up satellite (291.3 eV) as presented in Figure 4f [21]. Based on the results of Boehm titration, the concentration of oxygen-containing functional groups, including the carboxyl, Lactone, and Phenolic hydroxyl concentrations, are 0.414, 0.259, and 0.461 mmol·g^−1^, respectively. As reported, the oxygen-containing functional groups, e.g., O-C=O and C=O, on carbon-based materials can play a specific role in the catalytic degradation of organic contaminants [30]. For the core level of the Fe2p XPS spectrum, as given in Figure 4e, the peaks centered at 712.0, 710.6, and 706.8 eV can be assigned to Fe(III), Fe(II), and Fe(0), respectively [31]. The peak intensity corresponding to Fe(0) is relatively weaker than that of Fe(III) and Fe(II), demonstrating that Fe(0) particles might be covered by iron oxide. As mentioned in the literature, the formation of a surface oxide layer could be caused by the infiltration of oxygen into the pore structure during the synthesis procedure [16].

In order to further analyze the crystalline phase of Fe components, the XRD pattern of Fe@MC is shown in Figure 4g. The prepared material was composed of two types of iron crystals, α-Fe° and Fe_2_O_3_·FeO, which is also confirmed by the Fe valence states based on the XPS analysis. To be specific, the Fe_2_O_3_·FeO is the dominant form with the diffraction peaks at 2*θ* of 35.42°, 37.05°, 43.05°, 53.39°, 56.94°, 62.52°, 70.92°, and 73.95° assigned, respectively, to (311), (222), (400), (422), (511), (440), (620), and (533) crystal planes (JCPDS # 19-0629). The weak diffraction peaks at 44.67°, 65.02°, and 82.33°, corresponding to (110), (200), and (211) planes of α-Fe^0^ standard card (JCPDS # 06-0696), evidenced the appearance of zero-valent iron. Based on the TEM analysis, the Fe@MC is enveloped by a carbon shell having a lattice spacing of about 0.335 nm, which can be assigned to the (002) graphite plane. However, the lattice spacing of the core structure of Fe@MC is 0.202 nm attributing to the (110) α-Fe^0^ plane, which is consistent with the XPS analysis. The coverage of Fe(0) by a porous carbon shell should be beneficial for the anti-oxidation of Fe(0) during the application process. As shown in Appendix A, the nitrogen adsorption/desorption isotherm of Fe@MC is type-IV with H4-type hysteresis, indicating the existence of mesopores in the Fe@MC. The average pore diameter is 3.47 nm, with the maximum pore centered at 29.47 nm. The specific surface area is up to 335.19 m^2^/g, higher than other previously reported iron oxides [32] and carbon materials [33]. This mesoporous structure combined with a high surface area could be expected to promote electron transfer, the mass transfer process, PS activation, and TBBPA decomposition.

#### 3.4.2. Identification of Free Radicals

In order to reveal the primary free radicals involved in the degradation process of TBBPA, free radical quenching experiments were carried out. Generally speaking, methanol (MeOH) is considered to be effective in removing hydroxyl (OH) and sulfate radicals (SO_4_^−^) [34]. *Tert*-butanol (TBA), *p*-benzoquinone (BQ), and *L*-histidine can be used as the quencher for hydroxyl radicals (OH), superoxide radical (O_2_^−^), and singlet oxygen radical (^1^O_2_), respectively. Potassium iodide (KI) is used as a unique quencher to further evaluate the role of OH and SO_4_^−^ [35] on the Fe@MC surface during the reaction. The dosage of these quenchers in this study was chosen as [quencher]/[PS] = 10:1 and [KI]/[PS] = 10–100:1. As shown in Figure 5a, the degradation efficiency of TBBPA using methanol as a radical quencher is significantly reduced. At the same time, the TBA has a relatively low inhibitory effect on the degradation efficiency of TBBPA. This shows that the SO_4_^−^ radical plays a decisive role in removing TBBPA, and OH can also be produced by excessive SO_4_^−^ [36]. After adding *p*-benzoquinone (*p*-BQ) and *L*-histidine, the degradation efficiency of TBBPA, respectively, drops to 84.3% and 76.9%, indicating that superoxide radicals (O_2_^−^) and singlet oxygen free radicals (^1^O_2_) are also involved in the degradation process of TBBPA. When KI is added in high dosage ([KI]/[PS] = 100:1), only 14.6% of TBBPA can be removed, indicating that SO_4_^−^ and OH were mainly formed on the surface of Fe@MC materials.

Electron paramagnetic resonance (EPR) was employed for the identification of free radicals involved in the degradation process by using 5,5-dimethyl-1-pyrroline N oxide (DMPO) and 2,2,6,6-tetramethylpiperidine (TEMP) as free radical traps. As shown in Figure 5c, the OH and SO_4_^−^ signals are not detected in the PS solution. However, in the spectrum of the Fe@MC/PS system, the signals of OH, SO_4_^−^ and O_2_^−^ can be observed, indicating that OH, SO_4_^−^ and O_2_^−^ are produced during the degradation of TBBPA.

#### 3.4.3. TBBPA Degradation Mechanism

The nano-zero-valent iron (nZVI)/PS system was used as the control to reveal the mechanism of rapid degradation of TBBPA by the Fe@MC/PS system. During the reaction, the concentration of Fe(II), Fe(III) ions, and bromide in the solution were measured, and the results are shown in Figure 6a. It could be seen that the concentrations of Fe(II) and Fe(t) ions in the nZVI solution are much higher than those in the Fe@MC solution. The concentrations of Fe(II), Fe(III), and Fe(t) ions in the nZVI solution could increase rapidly in the first few minutes and then remain unchanged. The release of Fe(II) and Fe(III)/Fe(t) ions of nZVI in the solution was pronounced during the reaction, while Fe(III) became the primary ion in the reaction system after 30 min. In comparison, although the Fe(II) ion concentration in the Fe@MC/PS system was much lower than that in the nZVI/PS system, the concentration of Fe(III) in the solution was not increased during the reaction, and Fe(II) almost became the only iron species in the solution after 30 min. This suggested that the PS in the reaction system is almost completely consumed [37]. The special core-shell structure should be responsible for the low concentration of Fe ions in the solution by inhibiting the leaching of the metal core. Therefore, the presence of Fe(II) should not be the critical factor for TBBPA degradation. In contrast, excessive Fe(II) will consume SO_4_^−^ and negatively affect the degradation of pollutants [21]. Therefore, after 30 min of reaction, the degradation efficiency of TBBPA by the Fe@MC/PS system was 1.92 times higher than that of the nZVI/PS system.

During the reaction, the presence of Fe(III) and Fe(0) in Fe@MC could continuously generate Fe(II) at the solid-water interface to activate PS effectively (Figure 7). More importantly, the porous graphene-like carbon shell can provide the transfer channels for the SO_4_^−^ and work as the electron-transfer media to form numerous micro-reaction zones on the surface of Fe@MC for the oxidation of TBBPA, achieving 94.9% degradation of TBBPA within 10 min.

#### 3.4.4. Intermediates and Degradation Pathways

Density functional theory (DFT) was used to predict the reaction sites on TBBPA molecules vulnerable to radical attack. As given in Appendix A, electrophilic attack sites, nucleophilic attack sites, and radical attack sites were predicted by the TBBPA electron density Laplacian value. The blue zone indicates that the Laplacian value is negative, an area where electrons gather. In contrast, the yellow zone indicates that the Laplacian value is positive, which is the electron emission region. Therefore, the regions with negative Laplacian values would reflect the bonding situation and the strength of chemical bonds. It can be seen from Appendix A that the chemical bonds of the bromine atoms and phenolic hydroxyl groups linked on TBBPA are more fragile than the carbon atoms on the benzene rings. Electrical attacks are more likely to occur in these locations. The number of charges on each atom in TBBPA is obtained by bard charge analysis; the results also show that the carbon atoms (C_1_ and C_13_) linked by phenolic hydroxyl groups having the most positive charge would be more vulnerable to be attacked by electron-rich groups [38]. Based on the experimental results (Appendix A) and DFT analysis, a potential degradation pathway of TBBPA in the Fe@MC/PS system is proposed as given in Figure 8. Active radicals attack TBBPA to induce TBBPA degradation through two main pathways: (1) substitution of Br atoms by debromination; and (2) central carbon (C_7_) cleavage and hydroxylation of TBBPA. In scheme (1), as the generated **·**OH attacks the C-Br bond, the C-Br bond is cleaved to generate the debromination diphenol A, dibromodiphenol A, monobromodiphenol A, and bisphenol A. After that, Bisphenol A could be attacked by OH groups to generate intermediate R1 and radical R2. The intermediate R2 is then hydroxylated to produce phenol, which finally leads to mineralization. In scheme (2), the main degradation pathways are hydroxylation and oxidative ring opening of TBBPA [39]. In TBBPA, the bromide ion is replaced by a hydroxyl group, followed by the breakage of the C_7_ bond, and then an oxidative ring-opening reaction occurs to generate CO_2_ and H_2_O.

### 3.5. Toxicity of the Reaction Solution

TBBPA has a reportedly significant inhibitory effect on zebrafish embryonic development, leading to embryonic developmental malformations and even death. The main manifestations are egg coagulation, pericardium swelling, and vertebral curvature. The LC 50 and EC 50 of 48 hpf are 4.30 mg·L^−1^ and 2.56 mg·L^−1^, respectively, and the LC 50 and EC 50 of 120 hpf are, respectively, 2.88 mg·L^−1^ and 1.03 mg·L^−1^ [40]. Therefore, it is necessary to study the toxicity of TBBPA and its degradation products. Zebrafish embryos were used to assess the toxicity of mixtures containing TBBPA and its degradation products. Figure 9a shows the number of survived zebrafish embryos in the TBBPA solution before and after degradation for 30 min, and Figure 8b shows the number of survived embryos in the TBBPA solution with different degradation times. Based on the data given in Figure 9a, the toxicity of TBBPA is significantly reduced after degradation, and the overall toxicity of the product is decreased with time.

As seen from Figure 9b, due to the complexity of intermediate degradation products of TBBPA, the comprehensive toxicity of the degradation solution for 2 min is almost equal to that of the original solution. It may be due to the generation of many macromolecular organics with higher toxicity in the initial degradation stage. However, after 30 min, with the further degradation of these macromolecular organics, the toxicity of the degradation products is significantly decreased. Therefore, sufficient mineralization is significant and necessary for minimizing the toxicity of TBBPA.

## 4. Conclusions

This work synthesized an iron-carbon core-shell nanoparticle (Fe@MC) with a core-shell structure through a one-step hydrothermal method. Due to its unique core-shell structure and the presence of a porous graphite-like carbon shell, the agglomeration and leaching of Fe ions could be effectively eliminated. Moreover, the carbon shell layer could work as the electron transfer media and provide more active sites for the PS activation to form numerous micro-reaction zones on the surface of Fe@MC. The prepared material could effectively activate PS in a wide pH range, achieving 94.9% of TBBPA within 30 min at pH 7. Based on the radical scavenger and ESR studies, SO_4_^−^, OH, and non-radical reactive species mainly contributed to the degradation of TBBPA. Combined with DFT calculation and LC/MS analysis, two main degradation pathways of TBBPA were proposed. Although the comprehensive toxicity of the TBBPA solution was found to be increased within the initial degradation stage, the toxicity of the degradation products can be significantly decreased after 30 min. Therefore, the prepared Fe@MC could be considered a superior catalyst for activating PS to degrade TBBPA from water rapidly.

## Figures and Tables

**Figure 1 nanomaterials-12-04483-f001:**
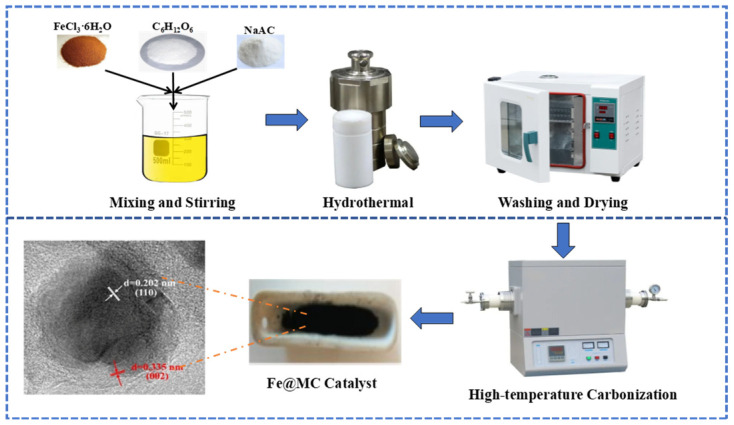
Catalyst preparation process.

**Figure 2 nanomaterials-12-04483-f002:**
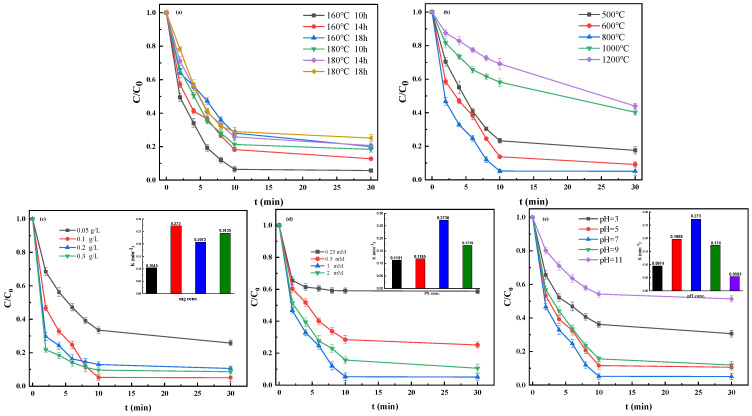
Effects of hydrothermal temperature, time (**a**), calcination temperature (**b**), Fe@MC dosages (**c**) ([PS] = 1 mM; [TBBPA] = 10 mg·L^−1^; pH = 7), PS dosages (**d**) (Fe@MC dosage = 0.1 g·L^−1^; [TBBPA] = 10 mg·L^−1^; pH = 7) and initial pH (**e**) (Fe@MC dosage = 0.1 g·L^−1^; [PS] = 1 mM; [TBBPA] = 10 mg·L^−1^) in the Fe@MC/PS system. (The internal illustrations show reaction rates under different conditions).

**Figure 3 nanomaterials-12-04483-f003:**
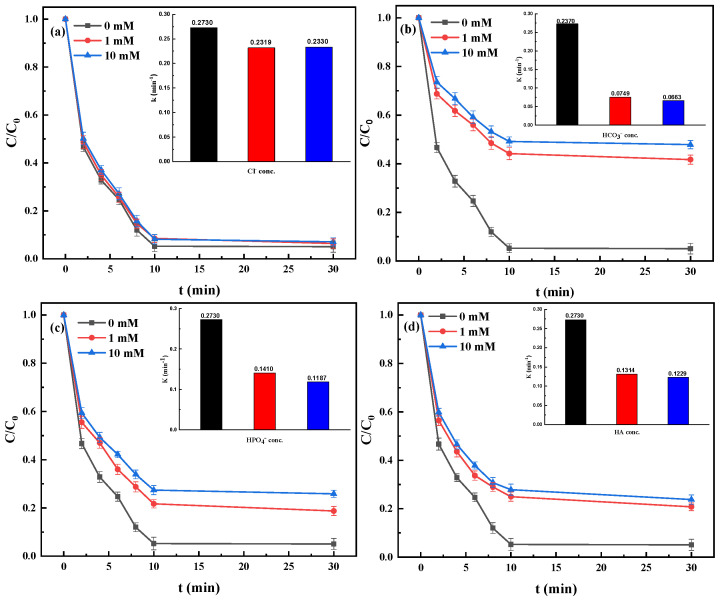
Effects of co-existing anions Cl^−^ (**a**), HCO_3_^−^ (**b**), H_2_PO_4_^−^ (**c**), and HA (**d**) on TBBPA removal performance of Fe@MC/PS system. (Fe@MC dosage = 0.1 g·L^−1^; [PS] = 1 mM; [TBBPA] = 10 mg·L^−1^ and the internal illustrations show reaction rates under different conditions).

**Figure 4 nanomaterials-12-04483-f004:**
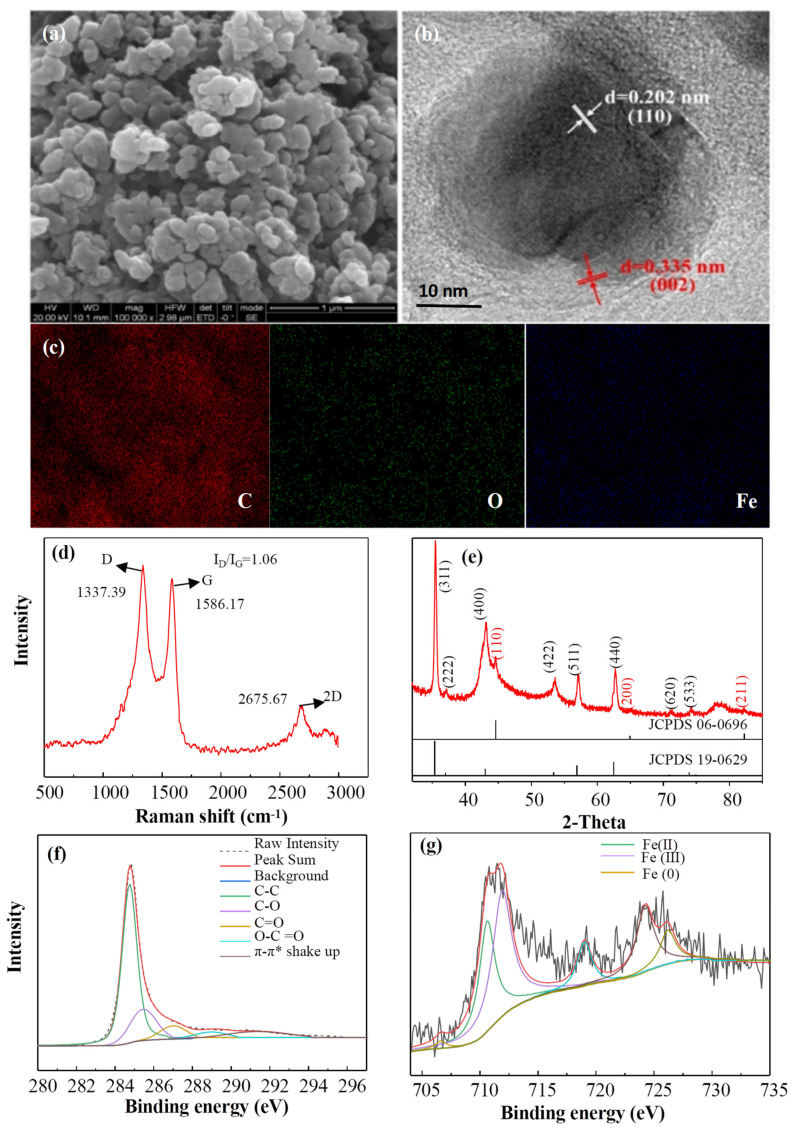
SEM (**a**), TEM (**b**), Elemental mapping (**c**), Raman spectrum (**d**), XRD pattern (**e**), C 1 s XPS spectrum (**f**) and Fe 2p XPS spectrum (**g**) images of the Fe@MC.

**Figure 5 nanomaterials-12-04483-f005:**
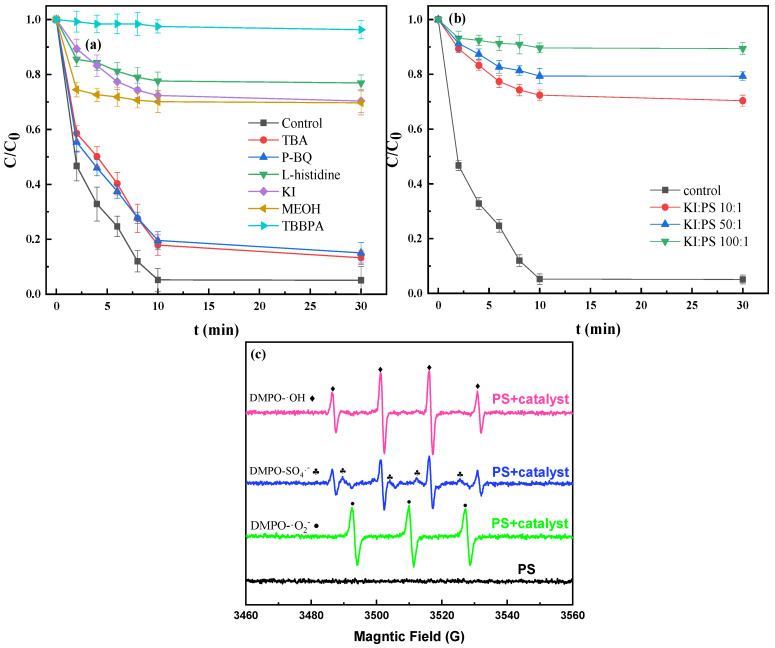
The effect of different quenchers (**a**) and concentrations of KI (**b**) on the performance of Fe@MC/PS system for removing TBBPA and ESR spectrum of reactive oxidative species (ROS) (using DMPO for SO_4_^−^, OH, TEMP for O_2_^−^) (**c**).

**Figure 6 nanomaterials-12-04483-f006:**
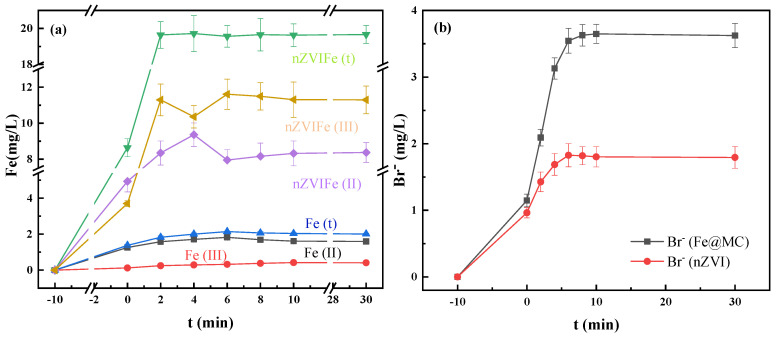
Concentrations of Fe(II), Fe(III) (**a**), and Br^−^ (**b**) during the reaction.

**Figure 7 nanomaterials-12-04483-f007:**
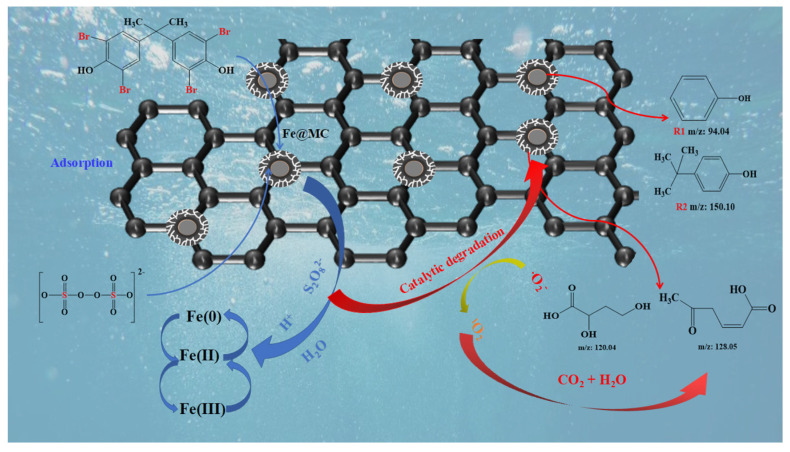
Degradation mechanism of TBBPA in the Fe@MC/PS system.

**Figure 8 nanomaterials-12-04483-f008:**
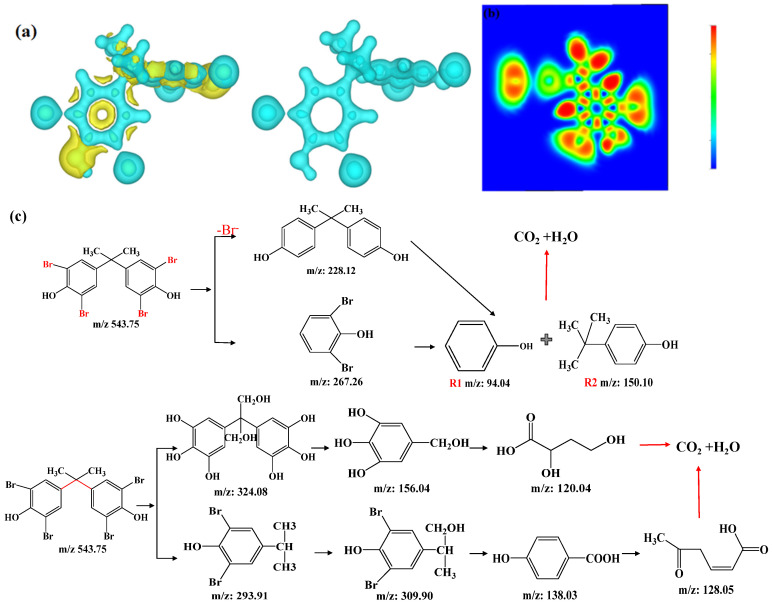
Electron density Laplacian analysis (**a**), projection map of electron localization function ELF (**b**) of TBBPA, and proposed TBBPA degradation pathways (**c**).

**Figure 9 nanomaterials-12-04483-f009:**
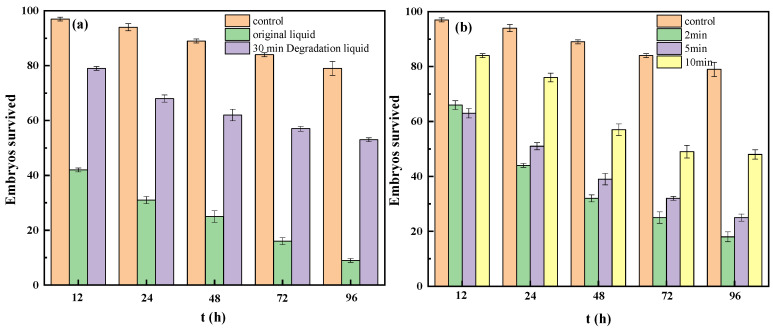
Survival numbers of zebrafish embryos in different solutions (**a**) and reaction time (**b**).

## Data Availability

The data presented in this study are available on request from the corresponding author.

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
