# Peer review of "One-Step Synthesized Iron-Carbon Core-Shell Nanoparticles to Activate Persulfate for Effective Degradation of Tetrabromobisphenol A: Performance and Activation Mechanism"

_nanomaterials, 2022, doi:10.3390/nano12244483_

Round 1
Reviewer 1 Report
This study reported the fabrication of core-shell iron-carbon composite nanoparticle (Fe@MC) using a one-step synthesis route for effectively tetrabromo-bisphenol A (TBBPA) degradation.
The goal of this study is good but there are many comments should be taken into consideration as follows:
1- In the Abstract section:
** The abstract part needs to be revised and modified to present only the most relevant objective and idea and to quickly get to the topic of the manuscript.
** Moreover, the novelty of the designed materials and the approach used need further explanations.
** Additionally, the Keywords should be revised for example the approach, as well as materials type "core-shell, Or....."
2- in the introduction part:
This part could be further improved by adding a few missing references, and some editing of the introduction as follows:
a- The author should focus on the exacerbation of the current research problem, the existing challenges, and the available solutions
b- More attention should be paid to the design of the materials used and the efforts made in the field related to the current study. In addition to highlighting the novelty of the current study.
c- The advantage of current material design needs further clarification
3- In the Experimental section
This part contains many steps of material design and application. In this regards, it is recommended to provided a schematic chart sufficient to explain all the steps to clarify this work and help the reader. This scheme should be designed in successive and clear steps with the addition of basic information synthesis (such as time, conc., ...) to achieve each step. In addition, part of the analysis (SEM, TEM) may be added to clarify the output, as well as sector of application.
4- In the results and discussion part:
a- First of all, it is recommended to start this section with a discussion of material preparation and analysis
b- In addition, the novelty of the prepared materials must be highlighted
c- The rules and mechanism of core-shell materials growth needs further discussion and clarification.
d- The (%) of each element should be added to the figure 3- EDS (c) analysis.
e- The mechanism of action needs more clarifications
** The role of the prepared materials in this process must be clearly highlighted
f- Is this work related to removing or degradation? "3.2. Removal optimization of TBBPA"
*** Therefore, the terminology used for this work should be checked and unified
g- In this context, the author called the synthetic material "iron-carbon composite nanoparticle (Fe@MC)", so does the author think it is better to use nanocomposite instead of composite nanoparticles?
** It is also recommended that the description of the prepared material be standardized to be consistent with the title of the work.
h- Real application is very important as a search goal. It is highly recommended to add a study related to the real application.
i- the figure captions is not enough to describe the figure, the author must provide more information and details enough to clarify the figures.
Author Response
请参阅附件。

Reviewer 2 Report
The manuscript submitted by Yu, et.al. for the degradation of TBBPA using Fe@MC materials provides enough invention and discussion, however authors need to consider the points mentioned below for the revision of manuscript.
1. Authors should elaborate the novelty of the work?
2. The Introduction section lacks the literature survey on Fe-based compounds for AOP, photocatalytic performances.
3. Please cross-check the symbols used in the manuscript. i.e., oC, should be corrected to o C.
4. Authors need to calculate the surface charge of the materials to support the description of line 179.
5. Regarding the XPS, discuss the polar and non-polar groups functionalities of Carbon and their associate impact on the degradation performance.
6. The kinetics study is not discussed in the manuscript. Also Fig. S8 is also not discussed?
7. The information provided in the supplementary files are not discussed or cited in the main manuscript.
8. The structure of the manuscript is different, application related information is provided in the first part of the manuscript however mid section describes the materials characteristics information.
9. The electrochemical study is not provided? EIS and Photocurrent study is missing.
10. Authors are requested to cross-check the symbols and grammatical errors in the manuscript.
11. What are the findings of the study?
Round 2
Reviewer 2 Report
Accept.
Author Response
see attachement
